# Colour Changes during the Carbamazepine Oxidation by Photo-Fenton

**Natalia Villota** [1,*], **Cristian Ferreiro** [2], **Hussein Ahmad Qulatein** [3], **Jose María Lomas** [1], **Luis Miguel Camarero** [1] and **José Ignacio Lombraña** [2]

1   Department of Chemical and Environmental Engineering, Faculty of Engineering of Vitoria-Gasteiz, University of the Basque Country UPV/EHU, Nieves Cano 12, 01006 Vitoria-Gasteiz, Spain; josemaria.lomas@ehu.eus (J.M.L.); luismiguel.camarero@ehu.es (L.M.C.)
2   Department of Chemical Engineering, Faculty of Science and Technology, University of the Basque Country UPV/EHU, Barrio Sarriena s/n, 48940 Leioa, Spain; cristian.ferreiro@ehu.eus (C.F.); ji.lombrana@ehu.eus (J.I.L.)
3   Department of Chemical Engineering, Faculty of Engineering, Anadolu University, 26555 Eskisehir, Turkey; husseinqullatein@gmail.com
*   Correspondence: natalia.villota@ehu.eus; Tel.: +34-9450-13248

**Abstract:** The oxidation of aqueous solutions of carbamazepine is conducted using the Fenton reagent, combined with the photolytic action of a 150 W medium pressure UV lamp, operating at $T = 40\,°C$. The effect of acidity is analysed at an interval pH = 2.0–5.0, verifying that operating at pH = 5.0 promotes colour formation (Colour = 0.15 AU). The effect of iron is studied, finding that the colour of the water increases in a linear way, Colour = 0.05 + 0.0075 [Fe]$_0$. The oxidising action of hydrogen peroxide is tested, confirming that when operating with [H$_2$O$_2$]$_0$ = 2.0 mM, the maximum colour is generated (Colour$_{max}$ = 0.381 AU). The tint would be generated by the degradation of by-products of carbamazepine, which have chromophoric groups in their internal structure, such as oxo and dioxocarbazepines, which would produce tint along the first minutes of oxidation, while the formation of acridones would slowly induce colour in the water.

**Keywords:** acridone; carbamazepine; colour; oxo-carbamazepine; photo-Fenton

## 1. Introduction

The study of emerging pollutants in wastewater, as well as its treatment and elimination, are receiving great attention in recent times due to their presence in many kinds of waters and their possible repercussions on the environment [1]. In almost all wastewater of both urban and industrial origin, different emerging pollutants have been detected in variable concentrations, depending on the activities conducted in the original areas of such waters. Recently, several governments are beginning to limit the presence of some of them, based on the Directive 2013/39/EU of the European Parliament, as well as the Council of 12 August 2013 Amending Directives 2000/60/EC and 2008/105/EC [2], although the effects that they cause or their content in the environment are largely unknown.

The main source of entry into the environment for these pollutants is through unprocessed wastewater and effluents from wastewater treatment plants (WWTPs). Conventional plants are not designed for the elimination of this type of micro-pollutants, so their removal in many cases is not complete. Based on this approach, a need arises for these studies, which seek to know the behaviour of emerging pollutants, which are selected based on European guidelines to be analysed in WWTPs. In this way, the aim of this work is to establish indicators of contamination throughout the different phases that form the treatment systems of these plants, being a key aspect to consider the degree of elimination of these contaminants in the different treatment processes currently used.

Among these priority substances, pharmaceutical products, being active biological substances, can affect living organisms in water even in small concentration. Pharmaceu-

ticals such as hormones, pain relievers, and antidepressants can have adverse influence on fish, crustaceans, and algae, because they have a similar kind of receptors as humans. The consequences on animals and plants can be very different from the pharmacological effects expected in humans. For this reason, there is a current need to develop new analysis methods that ensure the effectiveness of the AOPs, in order to conduct a correct design of the new processes [3].

Following the indications of Directive 2013/39/EU of the European Parliament, this work is part of a central line of research that is focussed on the development of techniques that allow the degradation of drugs, because there are resistant micro-pollutants contained in wastewater. The purpose is to prevent their transmission to water distribution networks based on the Commission Implementing Rule (EU) 2018/840 of 5 June 2018 [4].

This work focusses on the study of the degradation of the drug carbamazepine. This drug has been selected as a model pollutant of the study, due to its persistence in conventional treatment plants, as well as its wide presence in urban water [5]. Carbamazepine (CBZ) is a medicine utilised to treat neurological conditions such as epilepsy, depression, or bipolar disorder. In humans, around 72% is absorbed and metabolised in the liver, and 28% is excreted in feces. CBZ is one of the most frequently detected pharmaceutical compounds in urban aqueous systems [6,7]. On the other hand, the main metabolites detected in urine are BBZ-epoxide, CBZ-diol, CBZ-acridan, 2-OH-CBZ, and 3-OH-CBZ [5,8,9]. CBZ is a recalcitrant pollutant identified in the effluents of sewage treatment plants and in superficial waters, which has a potential impact on the environment due to its physico-chemical properties, since it is seldom eliminated in conventional water treatments [10].

Due to its potential effect on aquatic microorganisms and human health, there is a notable concern about its removal from water. Studies performed in the presence of CBZ in relevant concentrations show that it can induce disorders in lipid metabolism, as well as damage to mitochondria and DNA in fish [11,12]. Moreover, research published by Faisal et al. [5] shows that CBZ residues in drinking water could cause congenital malformations and/or neurological development problems after long-term intrauterine exposure or breastfeeding. On the other hand, analysis of UV-irradiated aqueous CBZ solutions reveals that acridine, a compound known to be carcinogenic, is one of the by-products formed [13].

Within this context, Advanced Oxidation Processes (AOPs) are presented as an alternative with great potential to effectively eliminate emerging pollutants. To perform the industrial implementation of AOPs, it is necessary to evaluate the different technologies to minimise toxic risks to human health [14], and to solve problems regarding technical feasibility, cost-effectiveness, and their own sustainability [15]. On the other hand, the low concentration levels in which these micro-pollutants are found in the waters limit the effectiveness of these treatments [16]. Assessing the effects induced by the discharge of these wastewaters into natural channels is a challenge, since it presents the difficulty of identifying numerous pollutants, metabolites, and transformation products in very low concentrations.

Among these technologies, this work tries to test the use of hydrogen peroxide combined with iron salts and ultraviolet (UV) light, called photo-Fenton Technology, in order to study the degradation of carbamazepine in aqueous solution. Ultraviolet light is a germicide emission that does not present any residual or secondary effects. Therefore, this technique has a great potential to become a useful tool with high viability. Nevertheless, it is necessary to develop a solid foundation of knowledge in the design of feasible processes for the degradation of emerging pollutants, which requires exhaustive research on the laboratory scale and in pilot plants.

## 2. Results

### 2.1. Colour Changes during Carbamazepine Oxidation

Figure 1 displays the colour changes that occur in the aqueous solution during the degradation of carbamazepine using the photo-Fenton process. The operating conditions in

the tests shown in Figure 1a lead to the formation of a tinted aqueous residue recalcitrant to oxidation. For this reason, it is chosen as a representative essay to analyse this phenomenon. The degradation of carbamazepine occurs during the first two hours of reaction following second-order kinetic guidelines. The generation of tint in the water occurs during the first 40 min of reaction until it reaches a maximum value that remains stable over time.

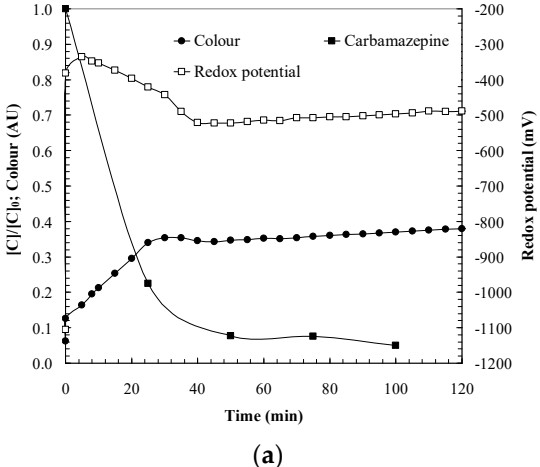

(**a**)

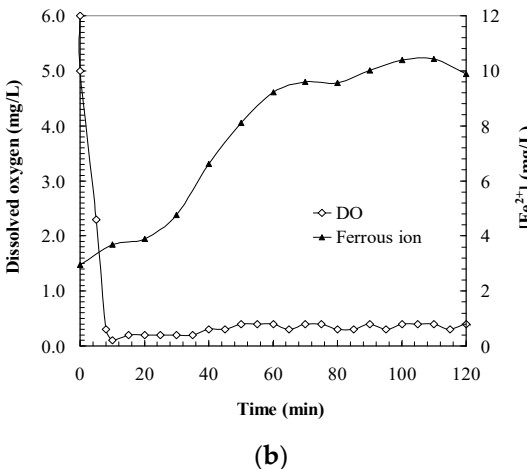

(**b**)

**Figure 1.** Water quality parameters analysed during carbamazepine oxidation by photo-Fenton: (**a**) Carbamazepine concentration, colour and redox potential. (**b**) Dissolved oxygen and ferrous ion. Experimental conditions: $[CBZ]_0 = 50.0$ mg/L; pH = 3.0; $[H_2O_2]_0 = 2.0$ mM; $[Fe]_0 = 10.0$ mg/L; [UV] = 150 W; T = 40 °C.

Analysing the redox potential values, an intense increase is observed during the first 5 min of the reaction until reaching a maximum value that decreases, arriving to a steady state after 40 min. This similar evolution between the colour and the redox potential changes makes it possible to associate the species that produce the hue changes in the water with the degradation intermediates of carbamazepine, which cause the redox potential values considered in the solution.

It should be noted that the increase in the redox potential during the first minutes of the reaction may be due to the oxidation of the ferrous ions to ferric, which is presented in Figure 1b. This allows verifying that approximately 70% of iron added to the reaction mixture in the form of ferrous ions is oxidised through the Fenton mechanism to ferric ions. Furthermore, during the course of the reaction, it is found that under the conditions tested, complete regeneration of the catalyst to ferrous ions occurs.

These results allow proposing a direct relationship between the redox potential and the reaction intermediates generated in the different stages of the carbamazepine oxidation mechanism. The substitution of groups of different nature (hydroxyl, oxo) in the aromatic rings affect the redox potential of the molecule, enlarging or reducing its value depending on the inducing effect of the substituent groups to accept or reject electrons in such a way that if the substitution in the ring is favored, they decrease the redox potential. In the case of hydroxylated carbamazepine molecules, when the aromatic ring loses the proton of the substituted hydroxyl group, electron delocalisation increases, thereby enlarging stability and causing the redox potential to decline [17]. Based on this hypothesis, it could be considered that the diminishment in redox potential would be related to the maximum concentration of dihydroxylated carbamazepines in the reaction medium, which would be contemplated as the precursor species of colour formation in water.

On the other hand, Figure 1b shows the evolution of dissolved oxygen (DO, mg $O_2$/L). During the first 10 min of the reaction, there is a high consumption of oxygen dissolved in water, until reaching levels around (DO = 0.1 mg $O_2$/L). This utilisation can be related to the oxidation process through the formation of strongly oxidising radical species. In this way, a highly oxidising environment is created that requires a large consumption of oxidising species. In addition, it is found that the moment when almost all the DO is

exhausted corresponds to the highest redox potential. This aspect can be associated to the maximum concentration of ferric ions generated in the Fenton reaction.

Next, the DO concentration begins to increase slightly until reaching levels of about 0.4 mg $O_2$/L after two hours of reaction. This behaviour is similar to that observed in studies reported in the bibliography during the oxidation of other organic pollutants [18], where this second stage of DO production presents a clear dependence on the nature of the oxidised species. In general, it is found that DO release is higher during the oxidation of organic matter that does not form organometallic complexes with iron, due to their molecular structure configuration. When the release of DO in the water is very slow, it is attributed to the fact that the degradation intermediates can form supramolecular structures with the ferric ions, preventing the iron regeneration.

In the case of the oxidation of carbamazepine shown in Figure 1b, it is observed that the DO release rate in water is very low ($k_{DO}$ = 0.0017 mg $O_2$/L min), although the ferric ions are completely regenerated to ferrous. This result could be attributed to oxygen evolution reactions, where free radicals participate. The conditions that facilitate the formation of tint in the water are related to the use of scarce oxidant with respect to the contaminant load. This leads to partial oxidation of carbamazepine towards the formation of colour precursor intermediates. By conducting the reaction with a shortage of oxidant, it causes the generated radical load to be consumed through the processes of oxidation of organic matter and iron regeneration. As a result, the interradical reactions producing oxygen release in the water are relegated.

## 2.2. pH Effect

Figure 2 presents the effect of pH on water colour changes during the oxidation of aqueous carbamazepine solutions, operating between pH = 2.0 and 5.0. It should be noted that the acidity has remained stable throughout the reaction at the initial established value. In the tests conducted, it was found that during the first 5 min of the oxidation, tint was generated in the water until it reached a maximum value and then decreases to a stable value, around 30 min of reaction time. PH determines the value of the colour area as well as the residual hue of the oxidised water. On the other hand, it is observed that operating between pH = 2.0 and 3.5, the maximum colour formation occurs at around 5 min of reaction. However, at pH = 4.0 and 5.0 the maximun colour formation occurs between 10 and 15 min.

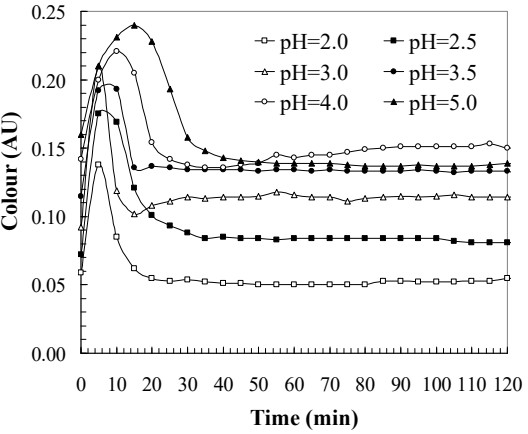

**Figure 2.** pH effect on colour changes in a photo-Fenton system during the carbamazepine oxidation. Experimental conditions: [CBZ]$_0$ = 50.0 mg/L; [H$_2$O$_2$]$_0$ = 15.0 mM; [Fe]$_0$ = 10.0 mg/L; [UV] = 150 W; T = 40 °C.

To analyse this result in more detail, Figure 3a represents the colour of the treated water once it has reached a steady state, together with the redox potential values. It is observed that both variables show a similar evolution regarding pH effect. By increasing the value from pH = 2.0 to 5.0, the intensity of the colour and the redox potential increases,

showing a maximum when carrying out the tests at pH = 5.0. As this pH increases, the colour and redox potential of the water decrease.

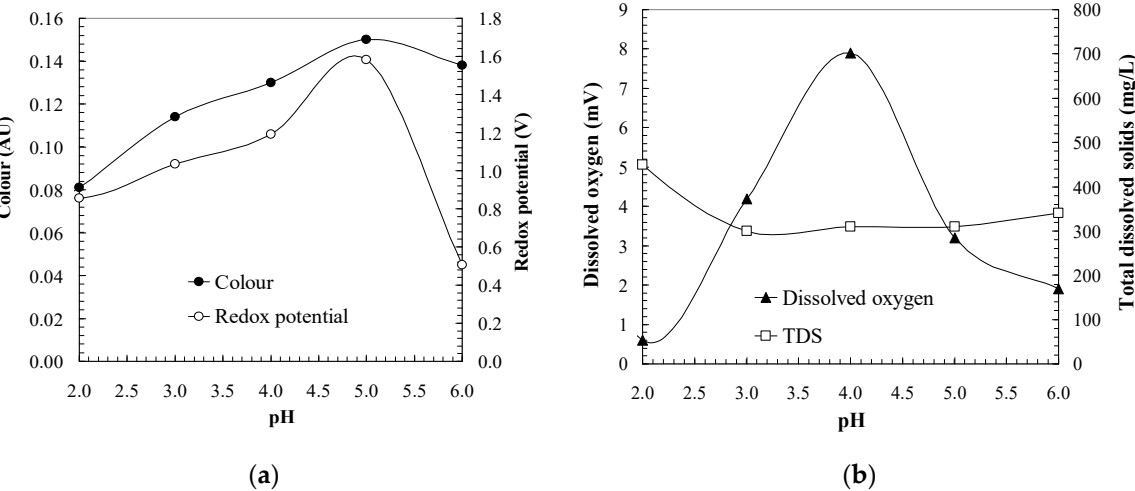

**Figure 3.** Indicator parameters of water quality analysed at the steady state: (**a**) Colour and redox potential. (**b**) Dissolved oxygen and total dissolved solids. Experimental conditions: $[CBZ]_0$ = 50.0 mg/L; $[H_2O_2]_0$ = 15.0 mM; $[Fe]_0$ = 10.0 mg/L; [UV] = 150 W; T = 40 °C.

To explain this effect, the speciation diagram of Fe (III) species as pH function in a photo-Fenton system [19] has been analysed. Then, it is found that the formation of the $Fe(OH)_2^-$ species in a photo-Fenton system potentially increases from pH = 2.0 until reaching its maximum at pH = 5.5. Thus, the effect of pH on colour formation could be associated with the presence of ferric hydroxide in the aqueous medium. The colour reduction operating at values higher than pH = 5.5 may be due to the fact that from this value, the formation of ferric hydroxide takes place, which would precipitate. This could cause a decrease in the concentration of iron dissolved, diminishing the aqueous tint.

Figure 3b displays the effect of pH on the concentration of DO in the water, which leads to verify a strong increase from pH = 2.0 to pH = 4.0, where the maximum concentration of DO occurs ([DO] = 7.9 mg $O_2$/L), and then, it decreases from pH = 4.0 to 6.0. This effect could be explained with the Pourbaix diagram for iron, which presents the predominance of the various chemical species in water for an element. Analysing the redox potential diagram of the medium as a function of pH, it can be verified that the experimental redox potential values measured for each pH (see Figure 3a) indicate that within the interval between pH = 2.0 and 4.0, the iron would be in the $Fe^{3+}$ form. Meanwhile, the values analysed at pH = 5.0 would indicate that iron would be in the $FeO_4^{2-}$ form and at pH = 6.0 in the $Fe_2O_3$ form. This change in the nature of the iron species that would coexist in the system could be related with the reactions of oxygen release.

### 2.3. Effect of Hydrogen Peroxide Dosage

During the oxidation treatment of aqueous carbamazepine samples, it is found that the water acquires colour during the first 20 min of reaction (Figure 4a). It is verified that the intensity of the tint depends on the dose of oxidant used. The results present two clear trends in the kinetics of colour formation. On the one hand, operating with low concentrations of oxidant, around $[H_2O_2]_0$ = 2.0 mM, corresponding to stoichiometric ratios of 1 mol CBZ: 10 mol $H_2O_2$, tint is generated in the water according to a ratio of 0.0086 AU/min, until reaching its maximum intensity ($Colour_{max}$ = 0.353 AU) at 30 min of reaction. Subsequently, the hue continues increasing but much more slowly, following ratios of 0.0005 AU/min, until it arrives at the steady state ($Colour_\infty$ = 0.381 AU).

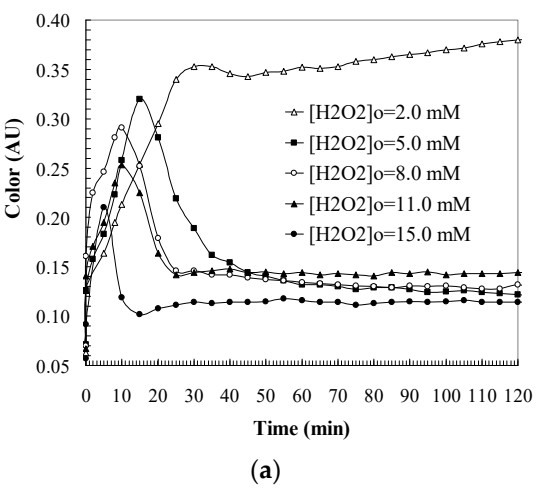

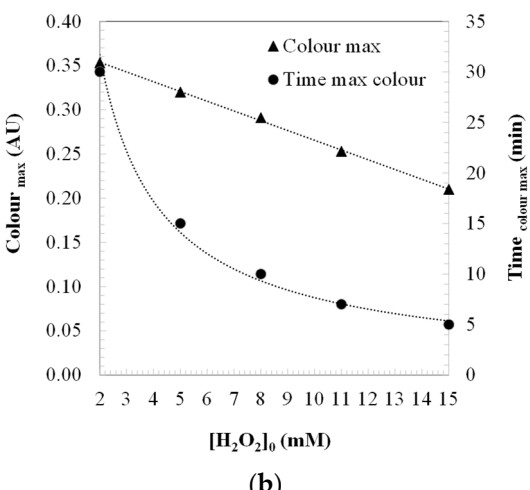

<div align="center">(<b>a</b>)            (<b>b</b>)</div>

**Figure 4.** (**a**) Effect of hydrogen peroxide on colour changes in a photo-Fenton system during the carbamazepine oxidation. (**b**) Maximum colour formation ($Colour_{max}$, AU) and time corresponding to the maximum colour formation ($Time_{colour\ max}$, min) as a function of the oxidant dosage. Experimental conditions: $[CBZ]_0 = 50.0$ mg/L; pH = 3.0; $[Fe]_0 = 10.0$ mg/L; [UV] = 150 W; T = 40 °C

Performing with oxidant concentrations greater than $[H_2O_2]_0 = 5.0$ mM, corresponding to stoichiometric ratios greater than 1 mol CBZ: 25 mol $H_2O_2$, the colour formation follows the evolution of a reaction intermediate, with rapid colour formation during the first minutes of oxidation, until reaching a maximum value, and decreasing until obtaining a colourless solution. The oxidant dosage determines both the maximum colour generated (Equation (1)) and the time in which the formation of the highest colour intensity occurs (Equation (2)), as it is shown in Figure 4b. This result indicates that the stoichiometric ratio of oxidant utilised determines the degree of oxidation achieved—that is, the stage of the carbamazepine degradation mechanism reached and, consequently, the nature of the intermediates that coexist in solution. As a result, the higher the molar ratio of oxidant, the lower the intensity of the tint generated, so that the formation of coloured species is reduced. The fact that under these conditions, a colourless oxidised residue is obtained shows that operating in all conditions, the dose of oxidant is sufficient to degrade the intermediates that provide tint to colourless species.

$$Colour_{max} = 0.3759 - 0.011\,[H_2O_2]_0 \quad (r^2 = 0.9988) \tag{1}$$

$$t_{colour\ max} = 58.31 \times [H_2O_2]_0^{-0.8813} \quad (r^2 = 0.9916) \tag{2}$$

The results shown indicate the existence of two stages in colour formation based on the carbamazepine degradation mechanism proposed in Figure 5. The first step takes place during the first stages of decomposition and leads to the formation of highly tinted species. This stage would involve hydroxylation reactions through the electrophilic attack of the hydroxyl radicals to the olefinic double bond in the central and lateral heterocyclic rings of carbamazepine, conducting to the formation of the corresponding hydroxylated carbamazepines. The action of hydroxyl radicals can generate a new hydroxylation of the molecule, leading to the creation of cis and trans-dihydroxy-carbamazepine [20]. The formation of the rare cis isomer appears to be less than that of trans [21]. Finally, the oxidation of these species would produce colour precursors, oxo and dioxo-carbazepines (10-OH-CBZ, 9-OH-CBZ, EP-CBZ, OX-CBZ), due to the presence of chromophore groups in their molecular structure.

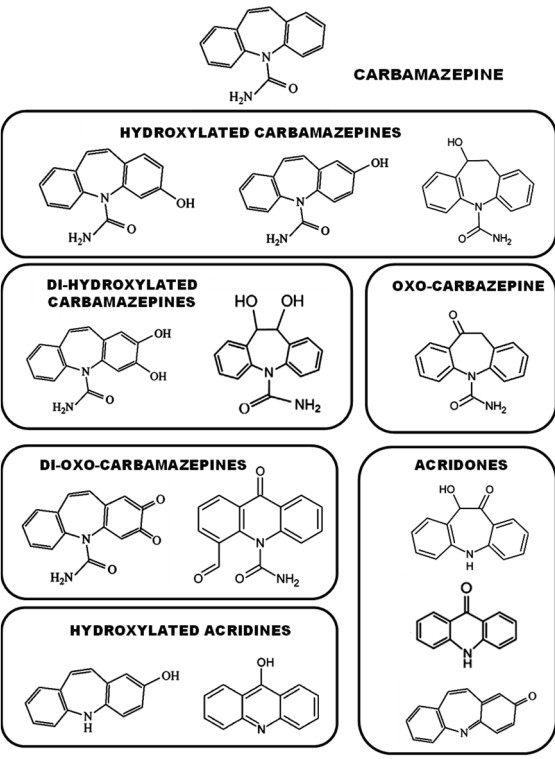

**Figure 5.** Reaction intermediates causing colour in oxidised carbamazepine solutions.

During the second stage, there would be the creation of additional species that coexist with those generated in the previous stage, which provide less intensity of tint to the water. In this case, it is possible to consider the formation of degradation by-products of the carbazepine species, generating hydroxylated molecules of acridine (9-OH-acridine) and the corresponding acridones that cause the additional contribution of colour.

Figure 6 shows the effect of the oxidant concentration used on several parameters that indicate the quality of the water once it is treated. Analysing the tint of the oxidised water, it is found that operating with concentrations $[H_2O_2]_0 = 2.0$ mM, the oxidation of carbamazepine leads to the formation of highly coloured species. On the other hand, working with concentrations higher than $[H_2O_2]_0 = 5.0$ mM, a colourless water is obtained. Simultaneously, the redox potential shows an evolution characterised by a slight decrease until reaching a minimum value ($[Redox]_{min} = -0.489$ V) in $[H_2O_2]_0 = 2.0$ mM, when the maximum colour formation take place (Colour$_{max} = 0.381$ AU). Subsequently, it increases practically linear with respect to the concentration of oxidant applied.

To explain this minimum value of redox potential, a relationship can be established between the evolution of the potential and the reaction intermediates generated in the different stages of the oxidation mechanism. Studies carried out on the effect of the substitution of groups of different nature in aromatic rings indicate that they affect the value of the redox potential of the molecule, increasing or decreasing depending on the inducing effect of the substituent groups to accept or transfer electrons [17]. Therefore, if ring substitution is favored, the redox potential value diminishes.

In the case of carbamazepine, there is a small stabilisation by resonance, which is attributable to electronic delocalisation. When the ring loses the proton of the substituted hydroxyl group, electron delocalisation increases, thus favoring stability and reducing the redox potential. Therefore, based on these hypotheses, the minimum value observed would be related to the maximum concentration of hydroxylated and dihydroxylated carbamazepines in the reaction medium, which would be the precursors of the tint that the solution acquires. By increasing the oxidant ratio, these intermediates are degraded, increasing the degree of oxidation, and it is found that the redox potential of the system evolves to positive values, which would indicate the formation of quinones and acridines.

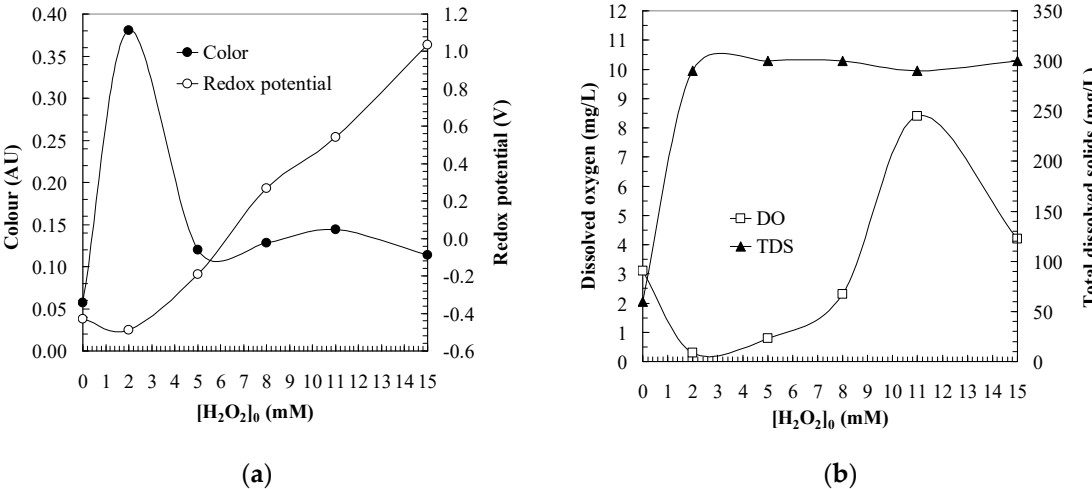

**Figure 6.** Indicator parameters of water quality analysed at the steady state: (**a**) Colour and redox potential. (**b**) Dissolved oxygen and total dissolved solids. Experimental conditions: $[CBZ]_0$ = 50.0 mg/L; pH = 3.0; $[Fe]_0$ = 10.0 mg/L; [UV] = 150 W; T = 40 °C.

The dissolved oxygen analysed in treated samples is consistent with their redox potential values. It is observed that the DO concentration in water increases as the treatment is conducted with higher concentrations of oxidant, up to a maximum operating point, which corresponds to $[H_2O_2]_0$ = 11.0 mM, with a DO = 8.4 mg $O_2$/L. However, in the test carried out using $[H_2O_2]_0$ = 15.0 mM, the DO experienced a big decrease until values of DO = 4.2 mg $O_2$/L. These lower levels of DO are observed throughout the course of the reaction, which could be due to operating with excess of oxidant with respect to the iron concentration. On the other hand, the concentration of Total Dissolved Solids (TDS, mg/L) remains constant in all the tests performed.

### 2.4. Effect of Iron Dosage

Figures 7 and 8 show the effect of catalyst concentration on the colour acquired by oxidised carbamazepine solutions. Operating with different iron concentrations (Figure 7a), it is observed that adding the iron dose established for each experiment increases tint to the initial carbamazepine solution ($Colour_0$, AU). The colour that the water gains shows a second degree polynomial increase (Equation (3)) with respect to the concentration of total iron supplied ($[Fe]_0$, mg/L). The initial iron added to the solution in the form of ferrous sulfate undergoes a series of equilibrium reactions between species, because the pH of the sample is adjusted to pH = 3.0 (Figure 7b). For this reason, one part of the iron ions is in a reduced state and the other is oxidised, being the ferric ions the providers of the tint to the water.

When the oxidant is added and the oxidation of the carbamazepine begins, the hue generated in the water increases until reaching a maximum value ($Colour_{max}$, AU) at 5 min after oxidation in all the tests conducted. This fact indicates that when using the same concentration of oxidant, the degradation intermediates of carbamazepine formed in water are similar species. Therefore, the colour peaks occur simultaneously, and following identical kinetics, they are displaced in parallel. This linear displacement is established by the iron concentration (Equation (4)).

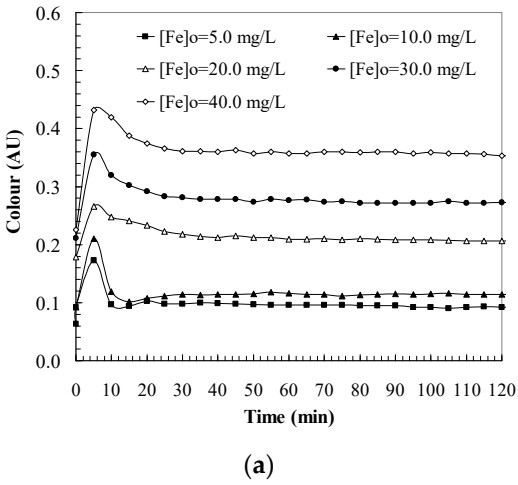 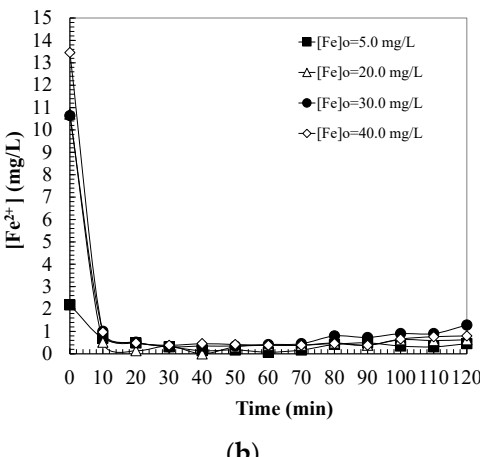

(**a**)  (**b**)

**Figure 7.** (**a**) Effect of iron on colour changes in a photo-Fenton system during the carbamazepine oxidation. (**b**) Ferrous ions concentration in water solution during carbamazepine oxidation. Experimental conditions: $[CBZ]_0$ = 50.0 mg/L; pH = 3.0; $[H_2O_2]_0$ = 15.0 mM; [UV] = 150 W; T = 40 °C.

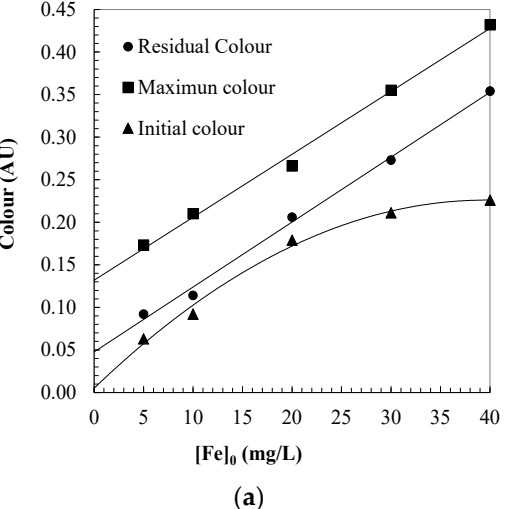 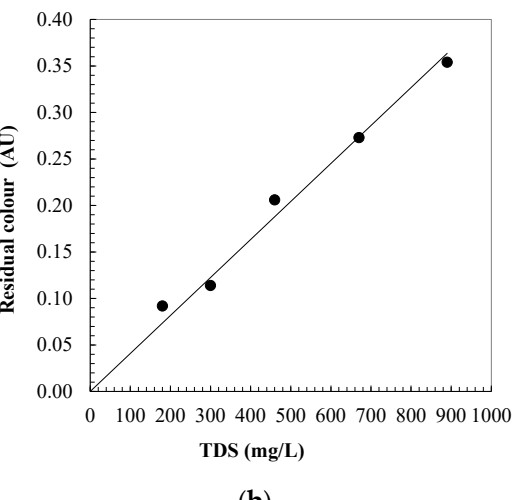

(**a**)  (**b**)

**Figure 8.** (**a**) Effect of iron dosage on water colour levels observed during the carbamazepine oxidation. (**b**) Relation-ship between total dissolved solids and the residual colour of water oxidized. Experimental conditions: $[CBZ]_0$ = 50.0 mg/L; pH = 3.0; $[H_2O_2]_0$ = 15.0 mM; [UV] = 150 W; T = 40 °C.

On the other hand, the persistant colour that lasts in the oxidised sample (Colour$_\infty$, AU) increases linearly with the iron concentration (Equation (5)). It is observed that both the maximum colour and the residual increase linearly with the total iron concentration, according to an average ratio of $k_{Fe}$ = 0.0075 AU L/mg Fe. Furthermore, it is found that they remain constant in all the tests: a difference between the maximum colour and the residual of 0.0843 AU. This tint value is explained by the contribution of iron species that can interact with the organic load of the water, forming metallic complexes, which are degraded during oxidation. As shown in Figure 8b, the lasting residual colour is provided by the iron species in suspension, which contribute linearly (Equation (7)) to the total suspended solids (TDS, mg/L).

$$Colour_0 = 0.0117\,[Fe]_0 - 0.0002\,[Fe]_0{}^2 \quad (r^2 = 0.9901) \tag{3}$$

$$Colour_{max} = 0.132 + 0.0074\,[Fe]_0 \quad (r^2 = 0.9946) \tag{4}$$

$$Colour_\infty = 0.0477 + 0.0076\,[Fe]_0 \quad (r^2 = 0.9961) \tag{5}$$

$$[TDS] = 72.982 + 20.211 \, [Fe]_0 \quad (r^2 = 0.9974) \tag{6}$$

$$Colour_\infty = 0.0004 \, [TDS] \quad (r^2 = 0.9826) \tag{7}$$

## 3. Materials and Methods

### 3.1. Experimental Methods

Samples of carbamazepine aqueous solutions ($[CBZ]_0$ = 50.0 mg/L, Fagron 99.1%) were studied in a photocatalytic 1.0 L reactor provided with an UV-150 W mercury lamp of medium pressure (Heraeus, 95%, transmission between 300 and 570 nm). Reactions started adding the iron catalyst as ferrous ion ($[Fe]_0$, mg/L), operating between $[Fe]_0$ = 5.0–40.0 mg/L ($FeSO_4$ 7 $H_2O$, Panreac 99.0%) and the oxidant dosage for each set of experiments, which varied between $[H_2O_2]_0$ = 0–15.0 mM (Panreac, 30% *w/v*). All the experiments were conducted at around 40 °C in order to simulate real working conditions, considering the heat absorbed by the water that is in direct contact with the UV lamp. Assays were performed under different initial pH conditions (pH between 2.0 and 5.0) in order to assess the effect of this parameter on colour formation during the oxidation of carbamazepine aqueous solutions. Acidity was kept constant adding NaOH and HCl.

### 3.2. Analytical Methods

Carbamazepine concentration (CBZ, mg/L) was assessed along the reaction at $\lambda = 210$ nm by a High-Performance Liquid Chromatograph attached to a spectrophotometer UV/Vis (HPLC Agilent 1200). Analysis was performed by injecting manually 20.0 µL samples, which were dragged by a carrier of 1.0 mL/min flow, consisting of a mixture of methanol and distilled water $MeOH/H_2O$: 80/20, through a Column $C_{18}$, XBridge Phenyl 5 µm $4.6 \times 250$ mm (Bridge Waters), with limit of detection 0.1 mg/L.

Colour expressed in Absorbance Units (AU) was quantified by the absorbance of the aqueous solution analysed at $\lambda = 455$ nm and ferrous ion ($[Fe^{2+}]$, mg/L) at $\lambda = 510$ nm by the phenanthroline method using an UV/Vis Spectrophotometer 930-Uvikon [22]. Dissolved oxygen (DO, mg/L) was measured by a DO-meter HI9142. Total dissolved solids (TDS, mg/L) were analysed by a TDS Metter Digital and Redox potential (V) by a conductimeter (Basic 20 Crison).

### 3.3. Liquid Chromatography-Mass Spectrometry to Elucidate the Intermediates of Carbamazepine Degradation

Samples were analysed by Liquid Chromatography-Mass Spectrometry to find the carbamazepine degradation pathways that induce high levels of colour in the water during the oxidation process. Analysis was performed with an LC/Q-TOF provided with an ionisation source ESI + Agilent Jet Stream, with the following conditions: Kinetex column EVO C18 ($100 \times 3$ mm) 2.6 µm. Moving phase 0.1% Formic Acid (A): Acetonitrile 0.1% Formic Acid (B). Gradient, %B: time (min): 20:0; 20:2; 100:24; 100:28; 20:30. Flow 0.3 mL/min. Column Temperature 35 °C. Injection volume 5 µL. Ionisation: Gas temperature 300 °C, drying gas 10 L/min, nebuliser 20 psig, shelf gas temperature 350 °C, shelf gas flow 11 L/min, frag 125 V. $V_{cap}$ 3500 V.

A screening method was developed, allowing the elution and ionisation of the majority of compounds in the sample. Before starting the analysis, the stabilisation of the system, the reproduction in the signals, and the correction of the exact masses were checked. With the aforementioned conditions, the chronogram shown in Figure 9 was attained.

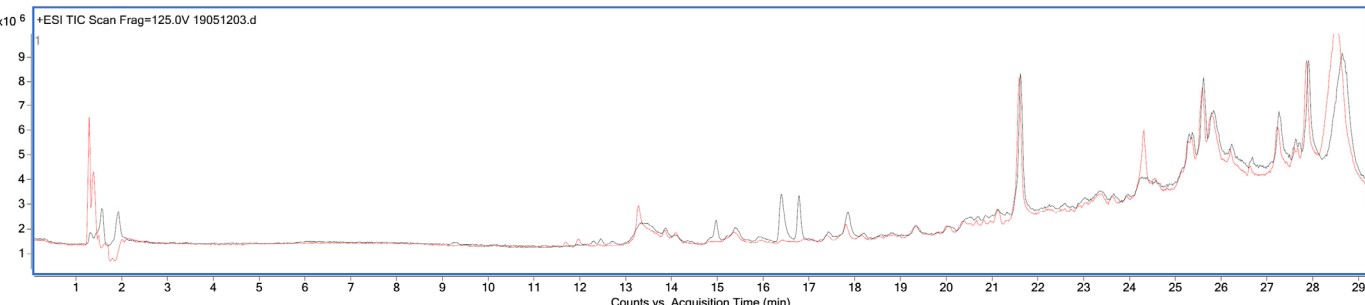

**Figure 9.** Chromatographic profile of a methanol blank (grey line) and of the sample (red line).

The search for compounds was performed using the Find deconvolution algorithm by molecular features and a subsequent screening of the proposed compounds, based on compounds detected in the blank, background noise, and minimum abundance of the compound (Figure 10). Appendix A summarises the major ions (*m/z*) and the experimental masses calculated for each of the compounds.

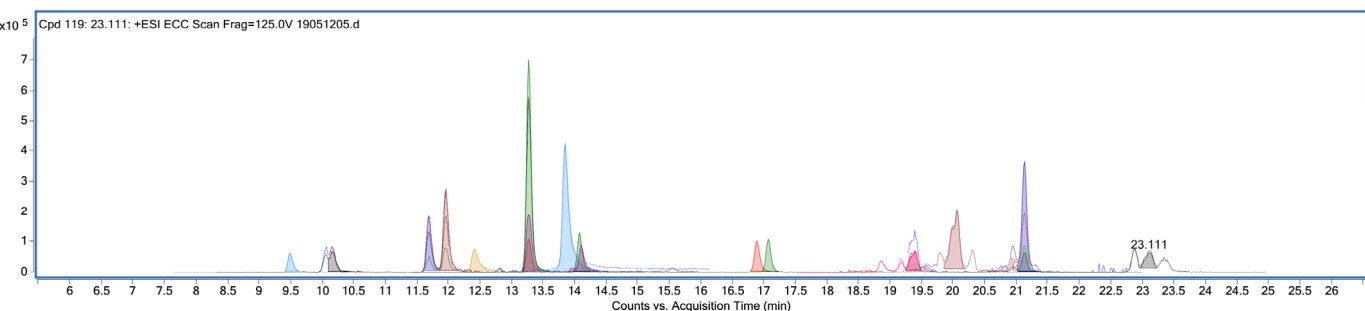

**Figure 10.** Chromatographic profile of the major compounds in the oxidised carbamazepine sample.

## 4. Conclusions

The stoichiometric ratio of oxidant determines the degree of oxidation achieved, that is, the nature of the intermediates that coexist in solution. Performing with low concentrations of oxidant, corresponding to stoichiometric ratios of 1 mol CBZ: 10 mol $H_2O_2$, colour is generated in the water until it reaches its maximum intensity (oxo and dioxo-carbazepines). Subsequently, the tint continues to increase more slowly, until arriving at the steady state, remaining a coloured aqueous residue that would contain hydroxylated acridines and acridones. Applying concentrations higher than 1 mol CBZ: 25 mol $H_2O_2$, the colour formation follows the evolution of a reaction intermediate, obtaining a colourless solution.

The initial iron added to the solution, in the form of ferrous sulfate, undergoes a series of equilibrium reactions between species. This is due to the fact that the acidity of the sample is adjusted to pH = 3.0 Therefore, a part of the iron ions are found in a reduced state and the another in its oxidised, being the ferric ions that provide tint to the water. Both the maximum colour and the persistent colour increase with the concentration of iron used in the treatment, according to an average ratio of $k_{Fe}$ = 0.0075 AU L/mg Fe. The maximum tint would be generated by the iron species that interact with the organic load, forming metallic complexes, while the lasting colour would be generated by the iron species in suspension.

**Author Contributions:** Conceptualization, N.V. and J.M.L.; methodology, L.M.C. and H.A.Q.; software, J.M.L; validation, C.F., J.I.L. and N.V.; formal analysis, J.I.L.; investigation, N.V., H.A.Q. and C.F.; resources, C.F.; data curation, N.V.; writing—original draft preparation, N.V., C.F. and H.A.Q.; writing—review and editing, J.M.L., L.M.C. and J.I.L.; visualization, N.V. and C.F.; supervision, L.M.C.; project administration, N.V. and J.I.L. and J.I.L. acquired the funding. All authors have read and agreed to the published version of the manuscript.

**Funding:** Authors are grateful to the University of the Basque Country UPV/EHU the financial support to carry out this research study through the scholarship Student Movility for Traineeships in the Erasmus + Programme between the Anadolu University in Eskisehir (Turkey) and the Faculty of Engineering Vitoria-Gasteiz (Spain), and the research Project PPGA20/33.

**Acknowledgments:** The authors thank for technical and human support provided by Central Service of Analysis from Álava—SGIker—UPV/EHU.

**Conflicts of Interest:** The authors declare no conflict of interest.

## Appendix A

| Label | RT | m/z | Mass | Height | Name | Score | Diff (DB, ppm) | Ions |
|---|---|---|---|---|---|---|---|---|
| Cpd 31: carbamazepina | 13,276 | 237,1029 | 236,0956 | 500597 | carbamazepina | 96,87 | -2,51 | 7 |
| Cpd 76: 13.857 | 13,857 | 274,2747 | 273,2674 | 352024 | | | | 3 |
| Cpd 112: 21.134 | 21,134 | 320,2563 | 297,2671 | 162272 | | | | 8 |
| Cpd 14: 10-OH-CBZ | 11,961 | 255,1133 | 254,106 | 157787 | 10-OH-CBZ | 97,82 | -2 | 5 |
| Cpd 33: 13.277 | 13,277 | 377,2089 | 376,2016 | 147084 | | | | 3 |
| Cpd 12: 10-OH-CBZ | 11,693 | 255,113 | 254,1057 | 111904 | 10-OH-CBZ | 99,74 | -0,69 | 4 |
| Cpd 104: 20.044 | 20,044 | 318,2412 | 295,2519 | 104589 | | | | 5 |
| Cpd 81: 14.083 | 14,083 | 237,1027 | 236,0954 | 93089 | | | | 3 |
| Cpd 30: 13.276 | 13,276 | 275,0584 | 274,0511 | 85219 | | | | 3 |
| Cpd 90: 16.894 | 16,894 | 238,11 | 237,1027 | 75562 | | | | 3 |
| Cpd 83: 14.110 | 14,11 | 318,2806 | 317,2733 | 69324 | | | | 3 |
| Cpd 115: 21.135 | 21,135 | 336,2303 | 335,223 | 65581 | | | | 3 |
| Cpd 17: 9-OH-acridine | 12,421 | 196,0759 | 195,0686 | 63672 | 9-OH-acridine | 99,55 | -1,06 | 2 |
| Cpd 91: 17.075 | 17,075 | 226,0865 | 225,0793 | 63218 | | | | 3 |
| Cpd 9: 10-OH-CBZ | 10,167 | 255,1132 | 254,1059 | 56756 | 10-OH-CBZ | 98,81 | -1,47 | 2 |
| Cpd 98: 19.375 | 19,375 | 280,2641 | 279,2569 | 55466 | | | | 2 |
| Cpd 6: 9.494 | 9,494 | 234,2067 | 233,1995 | 55020 | | | | 2 |
| Cpd 100: 19.384 | 19,384 | 320,2569 | 319,2497 | 54644 | | | | 2 |
| Cpd 119: 23.111 | 23,111 | 280,2638 | 279,2566 | 53960 | | | | 2 |
| Cpd 113: 21.134 | 21,134 | 356,3274 | 355,3201 | 50339 | | | | 2 |
| Cpd 86: 14.340 | 14,34 | 288,2895 | 287,2823 | 45926 | | | | 2 |
| Cpd 85: 14.155 | 14,155 | 210,0915 | 209,0842 | 42403 | | | | 2 |
| Cpd 35: 13.278 | 13,278 | 295,1557 | 294,1484 | 40689 | | | | 2 |
| Cpd 19: 12.963 | 12,963 | 267,1861 | 266,1788 | 38657 | | | | 2 |
| Cpd 111: 20.951 | 20,951 | 320,2565 | 297,2671 | 37049 | | | | 6 |
| Cpd 49: EP-CBZ | 13,392 | 253,0975 | 252,0902 | 36822 | EP-CBZ | 99,19 | -1,22 | 2 |
| Cpd 105: 20.049 | 20,049 | 334,215 | 333,2077 | 35819 | | | | 3 |
| Cpd 78: 14.068 | 14,068 | 290,2698 | 289,2625 | 35753 | | | | 2 |
| Cpd 16: 12.147 | 12,147 | 264,2324 | 263,2251 | 35155 | | | | 2 |
| Cpd 96: 19.177 | 19,177 | 280,264 | 279,2568 | 34358 | | | | 2 |
| Cpd 15: 12.028 | 12,028 | 246,2431 | 245,2358 | 33414 | | | | 2 |
| Cpd 107: 20.311 | 20,311 | 318,2406 | 295,2511 | 32304 | | | | 4 |
| Cpd 80: 14.082 | 14,082 | 275,0585 | 274,0513 | 30799 | | | | 3 |
| Cpd 29: 13.275 | 13,275 | 617,1398 | 1232,2651 | 30685 | | | | 3 |
| Cpd 27: 13.239 | 13,239 | 606,1349 | 1210,2552 | 30129 | | | | 3 |
| Cpd 106: 20.056 | 20,056 | 354,3127 | 353,3054 | 29989 | | | | 2 |
| Cpd 94: 18.871 | 18,871 | 280,2638 | 279,2566 | 29484 | | | | 2 |
| Cpd 25: 13.202 | 13,202 | 595,1277 | 1188,2408 | 28796 | | | | 3 |
| Cpd 10: OX-CBZ | 11,455 | 253,0974 | 252,0902 | 28240 | OX-CBZ | 99,39 | -1,06 | 2 |
| Cpd 79: 14.082 | 14,082 | 220,0761 | 219,0688 | 27418 | | | | 2 |
| Cpd 92: 18.868 | 18,868 | 320,2569 | 319,2496 | 27350 | | | | 2 |
| Cpd 38: 13.311 | 13,311 | 628,147 | 1254,2795 | 26627 | | | | 3 |
| Cpd 103: 19.794 | 19,794 | 318,2408 | 295,2515 | 26475 | | | | 4 |
| Cpd 28: 13.240 | 13,24 | 605,8839 | 1209,7533 | 26382 | | | | 3 |
| Cpd 60: 13.472 | 13,472 | 550,151 | 549,1437 | 26184 | | | | 2 |
| Cpd 26: 13.203 | 13,203 | 594,8776 | 1187,7406 | 26183 | | | | 3 |
| Cpd 89: 15.304 | 15,304 | 158,1537 | 157,1464 | 25035 | | | | 2 |
| Cpd 50: 13.409 | 13,409 | 532,5398 | 531,5325 | 24788 | | | | 2 |
| Cpd 82: 14.089 | 14,089 | 301,2856 | 300,2783 | 24337 | | | | 2 |
| Cpd 84: 14.126 | 14,126 | 239,118 | 238,1107 | 24171 | | | | 2 |
| Cpd 57: 13.443 | 13,443 | 541,3456 | 540,3383 | 24099 | | | | 2 |
| Cpd 75: EP-CBZ | 13,706 | 253,097 | 252,0898 | 23905 | EP-CBZ | 99,85 | 0,53 | 2 |
| Cpd 24: 13.164 | 13,164 | 584,1215 | 1166,2284 | 23833 | | | | 3 |
| Cpd 63: 13.503 | 13,503 | 558,9567 | 557,9494 | 23484 | | | | 2 |
| Cpd 87: 14.414 | 14,414 | 244,2638 | 243,2565 | 23224 | | | | 2 |
| Cpd 67: 13.533 | 13,533 | 567,762 | 566,7547 | 23093 | | | | 2 |
| Cpd 32: 13.277 | 13,277 | 616,8904 | 1231,7663 | 22863 | | | | 3 |

| Label | RT | m/z | Mass | Height | | | | Ions |
|---|---|---|---|---|---|---|---|---|
| Cpd 39: 13.312 | 13,312 | 627,8961 | 1253,7777 | 22613 | | | | 3 |
| Cpd 99: 19.383 | 19,383 | 336,2311 | 335,2238 | 22451 | | | | 3 |
| Cpd 41: 13.343 | 13,343 | 639,1532 | 1276,2919 | 22192 | | | | 3 |
| Cpd 8: 10.079 | 10,079 | 237,1025 | 236,0952 | 22176 | | | | 2 |
| Cpd 108: 20.338 | 20,338 | 293,2085 | 292,2012 | 22094 | | | | 3 |
| Cpd 47: 13.377 | 13,377 | 523,7354 | 522,7282 | 21838 | | | | 2 |
| Cpd 23: 13.163 | 13,163 | 583,871 | 1165,7274 | 21569 | | | | 3 |
| Cpd 18: 12.606 | 12,606 | 356,2799 | 355,2726 | 21417 | | | | 2 |
| Cpd 69: 13.562 | 13,562 | 576,5671 | 575,5598 | 21292 | | | | 2 |
| Cpd 4: 8.489 | 8,489 | 218,2118 | 217,2045 | 19862 | | | | 2 |
| Cpd 59: 13.471 | 13,471 | 550,3513 | 549,344 | 19427 | | | | 2 |
| Cpd 43: 13.361 | 13,361 | 263,0799 | 240,0905 | 19233 | | | | 4 |
| Cpd 48: 13.378 | 13,378 | 650,1607 | 1298,3068 | 19170 | | | | 3 |
| Cpd 110: 20.949 | 20,949 | 336,2306 | 335,2233 | 18835 | | | | 2 |
| Cpd 101: 19.392 | 19,392 | 316,2255 | 293,2362 | 18690 | | | | 3 |
| Cpd 70: 13.590 | 13,59 | 585,3719 | 584,3646 | 18102 | | | | 2 |
| Cpd 52: 13.410 | 13,41 | 532,7411 | 531,7338 | 17844 | | | | 2 |
| Cpd 61: 13.473 | 13,473 | 549,9508 | 548,9435 | 17777 | | | | 2 |
| Cpd 66: 13.531 | 13,531 | 567,9614 | 566,9542 | 17321 | | | | 2 |
| Cpd 22: 13.126 | 13,126 | 573,1151 | 1144,2156 | 17313 | | | | 2 |
| Cpd 42: 13.345 | 13,345 | 638,9041 | 1275,7937 | 17090 | | | | 3 |
| Cpd 55: 13.440 | 13,44 | 541,5456 | 540,5383 | 16747 | | | | 2 |
| Cpd 64: 13.503 | 13,503 | 559,1561 | 558,1488 | 16712 | | | | 2 |
| Cpd 62: 13.502 | 13,502 | 558,7568 | 557,7495 | 16704 | | | | 2 |
| Cpd 58: 13.444 | 13,444 | 541,1454 | 540,1382 | 16651 | | | | 2 |
| Cpd 7: 9.927 | 9,927 | 278,2328 | 277,2255 | 16503 | | | | 2 |
| Cpd 51: 13.410 | 13,41 | 661,1668 | 1320,319 | 16386 | | | | 2 |
| Cpd 53: 13.411 | 13,411 | 532,3405 | 531,3332 | 16342 | | | | 2 |
| Cpd 5: 9.370 | 9,37 | 262,2383 | 261,231 | 16255 | | | | 2 |
| Cpd 37: 13.309 | 13,309 | 505,923 | 504,9157 | 15944 | | | | 2 |
| Cpd 72: 13.615 | 13,615 | 594,1771 | 593,1699 | 15851 | | | | 2 |
| Cpd 36: 13.309 | 13,309 | 506,1242 | 505,1169 | 15539 | | | | 2 |
| Cpd 77: 14.042 | 14,042 | 288,2899 | 287,2826 | 15366 | | | | 2 |
| Cpd 46: 13.377 | 13,377 | 523,9355 | 522,9282 | 15213 | | | | 2 |
| Cpd 65: 13.530 | 13,53 | 567,5607 | 566,5534 | 14913 | | | | 2 |
| Cpd 44: 13.376 | 13,376 | 523,5355 | 522,5282 | 14899 | | | | 2 |
| Cpd 21: 13.125 | 13,125 | 572,8649 | 1143,7152 | 14659 | | | | 2 |
| Cpd 3: 5.852 | 5,852 | 120,0436 | 137,0473 | 14406 | | | | 2 |
| Cpd 2: 5.222 | 5,222 | 170,1176 | 187,1208 | 14396 | | | | 2 |
| Cpd 118: 22.908 | 22,908 | 336,3235 | 335,3163 | 14060 | | | | 2 |
| Cpd 45: 13.377 | 13,377 | 649,9099 | 1297,8052 | 13910 | | | | 3 |
| Cpd 73: 13.619 | 13,619 | 594,376 | 593,3687 | 13878 | | | | 2 |
| Cpd 68: 13.562 | 13,562 | 576,3662 | 575,3589 | 13754 | | | | 2 |
| Cpd 40: 13.343 | 13,343 | 514,7299 | 513,7226 | 13575 | | | | 2 |
| Cpd 34: 13.278 | 13,278 | 281,1401 | 280,1329 | 13193 | | | | 2 |
| Cpd 88: 14.862 | 14,862 | 224,1433 | 223,1361 | 13078 | | | | 2 |
| Cpd 102: 19.782 | 19,782 | 214,2174 | 213,2101 | 13071 | | | | 2 |
| Cpd 54: 13.412 | 13,412 | 660,9158 | 1319,8171 | 12993 | | | | 3 |
| Cpd 11: 11.546 | 11,546 | 239,0822 | 238,0749 | 12912 | | | | 2 |
| Cpd 1: 2.841 | 2,841 | 156,1019 | 173,1052 | 12774 | | | | 2 |
| Cpd 93: 18.869 | 18,869 | 336,2303 | 335,223 | 12673 | | | | 2 |
| Cpd 56: 13.443 | 13,443 | 672,1734 | 1342,3323 | 12590 | | | | 2 |
| Cpd 13: 11.958 | 11,958 | 220,0759 | 219,0686 | 12462 | | | | 2 |
| Cpd 97: 19.182 | 19,182 | 336,2305 | 335,2232 | 11979 | | | | 2 |
| Cpd 116: 21.404 | 21,404 | 431,1659 | 430,1586 | 11763 | | | | 2 |
| Cpd 71: 13.591 | 13,591 | 585,1719 | 584,1647 | 11328 | | | | 2 |
| Cpd 117: 22.123 | 22,123 | 429,3195 | 428,3122 | 11051 | | | | 2 |
| Cpd 74: 13.620 | 13,62 | 593,9763 | 592,969 | 10811 | | | | 2 |
| Cpd 95: 18.968 | 18,968 | 316,2252 | 293,2355 | 10142 | | | | 3 |
| Cpd 109: 20.479 | 20,479 | 488,3842 | 487,3769 | 10113 | | | | 2 |
| Cpd 20: 13.085 | 13,085 | 561,8572 | 1121,6999 | 10112 | | | | 2 |
| Cpd 114: 21.135 | 21,135 | 339,2996 | 338,2923 | 10052 | | | | 2 |

**Figure A1.** Major ions (*m/z*) and experimental masses calculated for each of the intermediate compounds detected in a sample of carbamazepine oxidized by photo-Fenton treatment under operating conditions that lead to the formation of coloured solution.

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
