# Peer review of "Colour Changes during the Carbamazepine Oxidation by Photo-Fenton"

_catalysts, doi:10.3390/catal11030386_

Round 1
Reviewer 1 Report
- Introduction can be more specific about studied problem instead of common ecological problems.
- Colour curve on fig.1a doesn't correspond with fig.2 and fig.4a.
- Not all the compounds, detected as products, were qualitatively chracterized.
Author Response
Journal
Catalysts (ISSN 2073-4344)
Manuscript ID
catalysts-1141316
Type
Article
Number of Pages
13
Title
Colour changes during the carbamazepine oxidation by photo-Fenton
Authors
Natalia Villota * , Cristian Ferreiro , Hussein A. Qulatein , Jose M. Lomas , Luis M. Camarero , J. Ignacio Lombraña
Abstract
The oxidation of aqueous solutions of carbamazepine is conducted using the Fenton reagent, com-bined with the photolytic action of a 150W medium pressure UV lamp, operating at T=40ºC. The effect of acidity is analyzed at an interval between pH=2.0-5.0, verifying that operating at pH=5.0 promotes colour formation (Colour=0.15 AU). The effect of iron is studied, founding that the col-our of the water increases in a linear way, Colour = 0.05 + 0.0075 [Fe]0. The oxidizing action of hydrogen peroxide is tested, confirming that when operating with [H2O2]0=2.0 mM the maximum colour is generated (0.381 AU). The tint would be generated by degradation of by-products of carbamazepine, which have chromophoric groups in their internal structure, such as oxo and di-oxocarbazepines, which would produce tint along the first minutes of oxidation, while the for-mation of acridones would slowly induce colour in the water.
Open Review
(x) I would not like to sign my review report
( ) I would like to sign my review report
English language and style
( ) Extensive editing of English language and style required
( ) Moderate English changes required
(x) English language and style are fine/minor spell check required
( ) I don't feel qualified to judge about the English language and style
|
Yes |
Can be improved |
Must be improved |
Not applicable |
|
|
Does the introduction provide sufficient background and include all relevant references? |
( ) |
(x) |
( ) |
( ) |
|
Is the research design appropriate? |
(x) |
( ) |
( ) |
( ) |
|
Are the methods adequately described? |
( ) |
(x) |
( ) |
( ) |
|
Are the results clearly presented? |
(x) |
( ) |
( ) |
( ) |
|
Are the conclusions supported by the results? |
(x) |
( ) |
( ) |
( ) |
Comments and Suggestions for Authors
- Introduction can be more specific about studied problem instead of common ecological problems.
Dear reviewer, we have restructured the text of the introduction to specify the objectives of the work.
We have removed the lines 36-42:
Recently, several governments are beginning to limit the presence of some of them, although the effects that they cause or their content in the environment are largely unknown (Directive 2013/39/EU of the European Parliament, as well as the Council of 12 August 2013 Amending Directives 2000/60/EC and 2008/105/EC, regarding priority substances in the field of water policy [2]). estates next: “Chemical contamination of surface waters exemplifies a threat to the aquatic environment, with effects such as acute and chronic toxicity in aquatic organisms, accumulation of pollutants in the ecosystem, loss of habitats and biodiversity. Moreover, it jeopardizes human health. As a priority, it is necessary to identify the causes of pollution and treat the emissions of pollutants at the source itself, in the most efficient way in economic and environmental aspects”.
We have removed the lines 52-55:
Following the indications of Directive 2013/39/EU of the European Parliament, about the state: “Wastewater processing can be very expensive. In order to facilitate affordable and more profitable treatment, the development of innovative water treatment technologies should be encouraged”. Therefore, this work is part of a central line of research focused on the development of techniques that allow the degradation of resistant micro-pollutants contained in wastewater.
We have reduced the lines 55-61:
The purpose is to prevent their transmission to water distribution networks, based on the Commission Implementing Rule (EU) 2018/840 of 5 June 2018. Consequently, it is established a watch list of substances for Union-wide monitoring in the field of water policy, pursuant to Directive 2008/105/EC of the European Parliament and the Council and repealing Commission Implementing Decision (EU) 2015/495 [3].
The part concerning carbamazepine (lines 89-105) has been anticipated and restructured.
- Colour curve on fig.1a doesn't correspond with fig.2 and fig.4a.
Thanks for your review. There is an error in the oxidant concentration data.
We have corrected the value of hydrogen peroxide concentration:
Figure 1. Water quality parameters analyzed during carbamazepine oxidation by photo-Fenton. Experimental conditions: [CBZ]0 = 50.0 mg / L; pH = 3.0; [H2O2]0 = 2.0 mM; [Fe]0 = 10.0 mg / L; [UV] = 150 W; T = 40 ºC.
- Not all the compounds, detected as products, were qualitatively chracterized.
In accordance with the data contrasted with the bibliography, only the compounds indicated in the paper have been identified.
Submission Date
24 February 2021
Date of this review
04 Mar 2021 18:18:03

Reviewer 2 Report
In this paper entitled “Colour changes during the carbamazepine oxidation by photo-Fenton”, Natalia Villota and co-workers investigated the oxidation of aqueous solutions of carbamazepine through the use of hydrogen peroxide in combination with iron salts and UV light, called photo-Fenton technology. Unfortunately, in my opinion the present manuscript shows severe issues, therefore at this stage is far from being worth of publication on the MDPI journal Catalysts.
First, I am unable to understand how COLOR is an physical quantity, i.e., a property that can be quantified by analytical measurement. I think it makes more sense using absorbance or transmittance of the aqueous solution, since they are well-defined and quantifiable physical quantities. Moreover, the mechanism of carbamazepine degradation into coloured species appeared quite unclear: I assume that each degradation product is characterized by its molar extinction coefficient: this is why I again believe that absorbance or transmittance are more suitable in this work.
Second, I believe that the introduction should be thoroughly revised. A) All EU Directive statements (lines 36-42 and 52-55) could be removed, by simply citing them in the bibliography. B) The content of the paragraph on lines 55-61 could also be drastically reduced. C) Moreover, I believe the part concerning carbamazepine (lines 89-105) could be anticipated, so that the final part of introduction concludes with the aim of the work. D) Concerning wastewater pollutants, authors focused the attention only on pharmaceutical products. Unfortunately, also textile industry, hair dye (DOI: 10.1016/j.toxlet.2007.05.108), leather and paper industries (DOI: 10.1016/S0376-7388(00)00399-9), and luminescent solar concentrator (LSC) technologies (DOI: 10.1002/slct.201800126) use a large amount of dyes potentially associated with water pollution. I believe that this point should be well specified at the beginning of the introduction, even with the support of the above mentioned literature.
Third, I invite the authors to carefully read the paper to check and correct various typos (for example, leave spaces before and after the = symbol).
Author Response
Journal
Catalysts (ISSN 2073-4344)
Manuscript ID
catalysts-1141316
Type
Article
Number of Pages
13
Title
Colour changes during the carbamazepine oxidation by photo-Fenton
Authors
Natalia Villota * , Cristian Ferreiro , Hussein A. Qulatein , Jose M. Lomas , Luis M. Camarero , J. Ignacio Lombraña
Abstract
The oxidation of aqueous solutions of carbamazepine is conducted using the Fenton reagent, com-bined with the photolytic action of a 150W medium pressure UV lamp, operating at T=40ºC. The effect of acidity is analyzed at an interval between pH=2.0-5.0, verifying that operating at pH=5.0 promotes colour formation (Colour=0.15 AU). The effect of iron is studied, founding that the col-our of the water increases in a linear way, Colour = 0.05 + 0.0075 [Fe]0. The oxidizing action of hydrogen peroxide is tested, confirming that when operating with [H2O2]0=2.0 mM the maximum colour is generated (0.381 AU). The tint would be generated by degradation of by-products of carbamazepine, which have chromophoric groups in their internal structure, such as oxo and di-oxocarbazepines, which would produce tint along the first minutes of oxidation, while the for-mation of acridones would slowly induce colour in the water.
Open Review
(x) I would not like to sign my review report
( ) I would like to sign my review report
English language and style
( ) Extensive editing of English language and style required
(x) Moderate English changes required
( ) English language and style are fine/minor spell check required
( ) I don't feel qualified to judge about the English language and style
|
Yes |
Can be improved |
Must be improved |
Not applicable |
|
|
Does the introduction provide sufficient background and include all relevant references? |
( ) |
( ) |
(x) |
( ) |
|
Is the research design appropriate? |
( ) |
( ) |
(x) |
( ) |
|
Are the methods adequately described? |
( ) |
(x) |
( ) |
( ) |
|
Are the results clearly presented? |
( ) |
( ) |
(x) |
( ) |
|
Are the conclusions supported by the results? |
( ) |
(x) |
( ) |
( ) |
Comments and Suggestions for Authors
In this paper entitled “Colour changes during the carbamazepine oxidation by photo-Fenton”, Natalia Villota and co-workers investigated the oxidation of aqueous solutions of carbamazepine through the use of hydrogen peroxide in combination with iron salts and UV light, called photo-Fenton technology. Unfortunately, in my opinion the present manuscript shows severe issues, therefore at this stage is far from being worth of publication on the MDPI journal Catalysts.
First, I am unable to understand how COLOR is an physical quantity, i.e., a property that can be quantified by analytical measurement. I think it makes more sense using absorbance or transmittance of the aqueous solution, since they are well-defined and quantifiable physical quantities. Moreover, the mechanism of carbamazepine degradation into coloured species appeared quite unclear: I assume that each degradation product is characterized by its molar extinction coefficient: this is why I again believe that absorbance or transmittance are more suitable in this work.
Dear reviewer
I agree with your comments. However, I want to clarify that in this work we use absorbance measurements, as we explained in the Experimental Methodology section. Therefore, when we say colour, we indicate in parentheses (AU), that is, Absorbance units. I am sorry that this fact may have created confusion for you. During the last fifteen years, our research group has developed an extensive analysis on the study of colour formation during the degradation of recalcitrant compounds, where we began to study the colour formation during the oxidation of phenol with Fenton reagent (Mijangos et al., 2006). Since then, we have always used the same nomenclature in all the publications to refer to the absorbance caused by the coloured species. Please, if you consider it appropriate, we would prefer to keep the same nomenclature formerly presented, in order to minimize confusion.
We have added in the text:
Colour expressed in Absorbance Units (AU) was quantified by the absorbance of the aqueous solution analyzed at λ=455 nm
Second, I believe that the introduction should be thoroughly revised.
- A) All EU Directive statements (lines 36-42 and 52-55) could be removed, by simply citing them in the bibliography.
Following your recommendation, we have removed the lines 36-42:
Recently, several governments are beginning to limit the presence of some of them, although the effects that they cause or their content in the environment are largely unknown (Directive 2013/39/EU of the European Parliament, as well as the Council of 12 August 2013 Amending Directives 2000/60/EC and 2008/105/EC, regarding priority substances in the field of water policy [2]). estates next: “Chemical contamination of surface waters exemplifies a threat to the aquatic environment, with effects such as acute and chronic toxicity in aquatic organisms, accumulation of pollutants in the ecosystem, loss of habitats and biodiversity. Moreover, it jeopardizes human health. As a priority, it is necessary to identify the causes of pollution and treat the emissions of pollutants at the source itself, in the most efficient way in economic and environmental aspects”.
Following your recommendation, we have removed the lines 52-55:
Following the indications of Directive 2013/39/EU of the European Parliament, about the state: “Wastewater processing can be very expensive. In order to facilitate affordable and more profitable treatment, the development of innovative water treatment technologies should be encouraged”. Therefore, this work is part of a central line of research focused on the development of techniques that allow the degradation of resistant micro-pollutants contained in wastewater.
- B) The content of the paragraph on lines 55-61 could also be drastically reduced.
Following your recommendation, we have reduced the lines 55-61:
The purpose is to prevent their transmission to water distribution networks, based on the Commission Implementing Rule (EU) 2018/840 of 5 June 2018. Consequently, it is established a watch list of substances for Union-wide monitoring in the field of water policy, pursuant to Directive 2008/105/EC of the European Parliament and the Council and repealing Commission Implementing Decision (EU) 2015/495 [3].
- C) Moreover, I believe the part concerning carbamazepine (lines 89-105) could be anticipated, so that the final part of introduction concludes with the aim of the work.
Following your recommendation, we have anticipated the lines 89-105.
- D) Concerning wastewater pollutants, authors focused the attention only on pharmaceutical products. Unfortunately, also textile industry, hair dye (DOI: 10.1016/j.toxlet.2007.05.108), leather and paper industries (DOI: 10.1016/S0376-7388(00)00399-9), and luminescent solar concentrator (LSC) technologies (DOI: 10.1002/slct.201800126) use a large amount of dyes potentially associated with water pollution. I believe that this point should be well specified at the beginning of the introduction, even with the support of the above mentioned literature.
In this regard, I want to emphasize that carbamazepine degradation intermediates are not pigments nor colourants. They are not compounds that tint water like textile or hair dyes. This study is not related to the references that you indicate. The intermediates identified in this work are brown and yellow compounds. So, when they are generated in solution, the water acquires the colour of the compound. Because carbamazepine is colourless, when it is oxidized to these substances, it is possible to detect them visually. Besides, another characteristic of these products is that they provide greater toxicity to water than carbamazepine itself. Consequently, colour can be used as an indicator parameter for the presence of these intermediates.
Third, I invite the authors to carefully read the paper to check and correct various typos (for example, leave spaces before and after the = symbol).
We have checked and corrected typos in all the manuscript.

Round 2
Reviewer 2 Report
In this revised version of entitled “Colour changes during the carbamazepine oxidation by photo-Fenton”, Natalia Villota and collaborators addressed quite satisfactorily all the issues listed in my previous report, thus allowing to improve the quality of the work. I believe that the present paper now meets the standards for publication in the MDPI Catalysts journal, thus suggesting its acceptance in the present form.